# HYPERPARAMETER OPTIMIZATION WITH HYPERNETS

**Jonathan Lorraine & David Duvenaud**
Department of Computer Science
University of Toronto
{lorraine, duvenaud}@cs.toronto.edu

## ABSTRACT

Machine learning models are often tuned by nesting optimization of model weights inside the optimization of hyperparameters. We give a method to collapse this nested optimization into joint stochastic optimization of weights and hyperparameters. Our process trains a neural network to output approximately optimal weights as a function of hyperparameters. We show that our technique converges to locally optimal weights and hyperparameters for sufficiently large hypernets. We compare this method to standard hyperparameter optimization strategies.

## 1 INTRODUCTION

Model selection and hyperparameter tuning is a significant bottleneck in designing predictive models. Hyperparameter optimization is a nested optimization: The inner optimization finds model parameters $w$ which minimize the training loss $\mathcal{L}_{\text{Train}}$ given hyperparameters $\lambda$. The outer optimization chooses $\lambda$ to reduce a validation loss $\mathcal{L}_{\text{Valid.}}$:

$$\underset{\lambda}{\text{argmin}} \ \underset{\text{Valid.}}{\mathcal{L}} \left( \underset{w}{\text{argmin}} \ \underset{\text{Train}}{\mathcal{L}} (w, \lambda) \right) \quad (1)$$

Usually, we estimate parameters with stochastic optimization, but the most likely parameters are a deterministic function of the hyperparameters $\lambda$:

$$w^*(\lambda) = \underset{w}{\text{argmin}} \ \underset{\text{Train}}{\mathcal{L}} (w, \lambda) \quad (2)$$

We propose to *learn this function*. Specifically, we train a neural network that takes hyperparameters as input, and outputs of an approximately optimal set of weights given the hyperparameters.

This formulation provides two major benefits: First, we can train the hypernet to convergence using stochastic gradient descent (SGD) without training any particular model to completion. Second, differentiating through the hypernet allows us to optimize hyperparameters with stochastic gradient-based optimization.

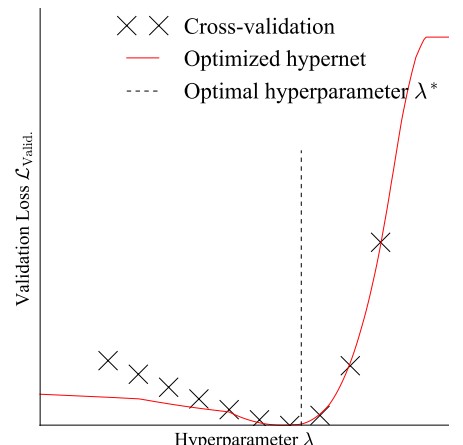

Figure 1: The validation loss of a neural net, estimated by cross-validation (crosses) or by a hypernet (line), which outputs $7,850$-dimensional network weights. Cross-validation requires optimizing from scratch each time. The hypernet can be used to evaluate the validation loss cheaply.

## 2 TRAINING A NETWORK TO OUTPUT OPTIMAL WEIGHTS

How can we teach a *hypernet* (Ha et al., 2016) to output approximately optimal weights to another neural network? The basic idea is that at each iteration, we ask a hypernet to output a set of weights given some hyperparameters: $w = w_\phi(\lambda)$. Instead of updating the weights $w$ using the training loss gradient $\partial \mathcal{L}(w)/\partial w$, we update the hypernet weights $\phi$ using the chain rule: $\frac{\partial \mathcal{L}(w_\phi)}{\partial w_\phi} \frac{\partial w_\phi}{\partial \phi}$. This expansion allows us to optimize the hyperparameters $\lambda$ with the validation loss gradient $\frac{\partial \mathcal{L}(w_\phi(\lambda))}{\partial w_\phi(\lambda)} \frac{\partial w_\phi(\lambda)}{\partial \lambda}$. We call this method *hyper-training* and contrast it with standard training methods.

| **Algorithm 1** Optimization of hypernet, then hyperparameters | **Algorithm 2** Joint optimization of hypernet and hyperparameters |
|---|---|
| initialize $\phi$, initialize $\hat{\lambda}$ | initialize $\phi, \hat{\lambda}$ |
| **loop** | **loop** |
| $\quad$ $\mathbf{x} \sim$ Training data, $\lambda \sim p(\lambda)$ | $\quad$ $\mathbf{x} \sim$ Training data, $\lambda \sim p(\lambda\|\hat{\lambda})$ |
| $\quad$ $\phi = \phi - \alpha \nabla_\phi \mathcal{L}_{\text{Train}}(\mathbf{x}, \mathrm{w}_\phi(\lambda), \lambda)$ | $\quad$ $\phi = \phi - \alpha \nabla_\phi \mathcal{L}_{\text{Train}}(\mathbf{x}, \mathrm{w}_\phi(\lambda), \lambda)$ |
| **loop** | |
| $\quad$ $\mathbf{x} \sim$ Validation data | $\quad$ $\mathbf{x} \sim$ Validation data |
| $\quad$ $\hat{\lambda} = \hat{\lambda} - \beta \nabla_{\hat{\lambda}} \mathcal{L}_{\text{Valid.}}(\mathbf{x}, \mathrm{w}_\phi(\hat{\lambda}))$ | $\quad$ $\hat{\lambda} = \hat{\lambda} - \beta \nabla_{\hat{\lambda}} \mathcal{L}_{\text{Valid.}}(\mathbf{x}, \mathrm{w}_\phi(\hat{\lambda}))$ |
| Return $\hat{\lambda}, \mathrm{w}_\phi(\hat{\lambda})$ | Return $\hat{\lambda}, \mathrm{w}_\phi(\hat{\lambda})$ |

A comparison of standard hyperparameter optimization, our first algorithm, and our joint algorithm.

Our method is closely related to the concurrent work of Brock et al. (2017), whose SMASH algorithm also approximates the optimal weights as a function of model architectures, to perform a gradient-free search over discrete model structures. Their work focuses on efficiently estimating the performance of a variety of model architectures, while we focus on efficiently exploring continuous spaces of models.

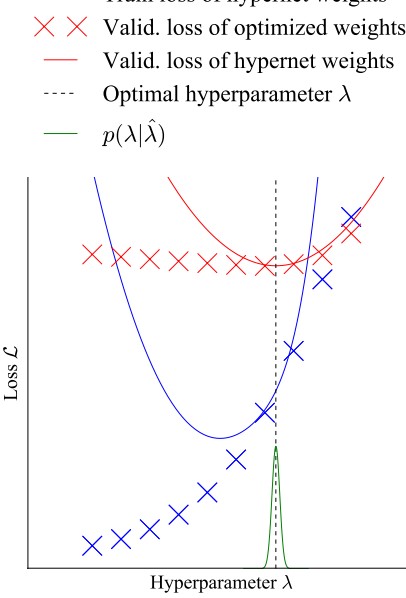

- ✕ ✕   Train loss of optimized weights
- ——   Train loss of hypernet weights
- ✕ ✕   Valid. loss of optimized weights
- ——   Valid. loss of hypernet weights
- ----   Optimal hyperparameter $\lambda$
- ——   $p(\lambda\|\hat{\lambda})$

Figure 2: Training and validation losses of a neural network, estimated by cross-validation (crosses) or a linear hypernet (lines). The hypernet's limited capacity makes it only accurate where the hyperparameter distribution puts mass.

### 2.1 ASYMPTOTIC CONVERGENCE PROPERTIES

This section proves that Algorithm 1 converges to a local best-response under mild assumptions.

**Theorem 2.1.** *Sufficiently powerful hypernets can learn continuous best-response functions, which minimizes the expected loss for all hyperparameter distributions with convex support.*

There exists $\phi^*$, such that for all $\lambda \in \text{support}(p(\lambda))$,

$$\mathcal{L}_{\text{Train}}(\mathrm{w}_{\phi^*}(\lambda), \lambda) = \min_{\mathrm{w}} \mathcal{L}_{\text{Train}}(\mathrm{w}, \lambda) \text{ and}$$

$$\phi^* = \operatorname*{argmin}_\phi \mathbb{E}_{p(\lambda')}\left[\mathcal{L}_{\text{Train}}(\mathrm{w}_\phi(\lambda'), \lambda')\right]$$

*Proof.* If $\mathrm{w}_\phi$ is a universal approximator (Hornik, 1991) and the best-response is continuous in $\lambda$ (which allows approximation by $\mathrm{w}_\phi$), then there exists optimal hypernet parameters $\phi^*$ such that for all hyperparameters $\lambda$, $\mathrm{w}_{\phi^*}(\lambda) = \operatorname{argmin}_{\mathrm{w}} \mathcal{L}_{\text{Train}}(\mathrm{w}, \lambda)$. Thus, $\mathcal{L}_{\text{Train}}(\mathrm{w}_{\phi^*}(\lambda), \lambda) = \min_{\mathrm{w}} \mathcal{L}_{\text{Train}}(\mathrm{w}, \lambda)$. In other words, universal approximator hypernets can learn continuous best-responses.

Substituting $\phi^*$ into the training loss gives $\mathbb{E}_{p(\lambda)}[\mathcal{L}_{\text{Train}}(\mathrm{w}_{\phi^*}(\lambda), \lambda)] = \mathbb{E}_{p(\lambda)}[\min_\phi \mathcal{L}_{\text{Train}}(\mathrm{w}_\phi(\lambda), \lambda)]$. By Jensen's inequality, $\min_\phi \mathbb{E}_{p(\lambda)}[\mathcal{L}_{\text{Train}}(\mathrm{w}_\phi(\lambda), \lambda)] \geq \mathbb{E}_{p(\lambda)}[\min_\phi \mathcal{L}_{\text{Train}}(\mathrm{w}_\phi(\lambda), \lambda)]$. To satisfy Jensen's requirements, we have $\min_\phi$ as our convex function on the convex vector space of functions $\{\mathcal{L}_{\text{Train}}(\mathrm{w}_\phi(\lambda), \lambda) \text{ for } \lambda \in \text{support}(p(\lambda))\}$. To guarantee convexity of the vector space we require that $\text{support}(p(\lambda))$ is convex and $\mathcal{L}_{\text{Train}}(\mathrm{w}, \lambda) = \mathbb{E}_{\mathbf{x} \sim \text{Train}}[\mathcal{L}_{\text{Pred}}(\mathbf{x}, \mathrm{w})] + \mathcal{L}_{\text{Reg}}(\mathrm{w}, \lambda)$ with $\mathcal{L}_{\text{Reg}}(\mathrm{w}, \lambda) = \lambda \cdot \mathcal{L}(\mathrm{w})$. Thus, $\phi^* = \operatorname{argmin}_\phi \mathbb{E}_{p(\lambda)}[\mathcal{L}_{\text{Train}}(\mathrm{w}_\phi(\lambda), \lambda)]$. In other words, if the hypernet learns the best-response it will simultaneously minimize the loss for every point in $\text{support}(p(\lambda))$. $\square$

Thus, having a universal approximator and a continuous best-response implies for all $\lambda \in \text{support}(p(\lambda))$, $\mathcal{L}_{\text{Valid.}}(\mathrm{w}_{\phi^*}(\lambda)) = \mathcal{L}_{\text{Valid.}}(\mathrm{w}^*(\lambda))$ because $\mathrm{w}_{\phi^*}(\lambda) = \mathrm{w}^*(\lambda)$.

## 2.2 Jointly training parameters and hyperparameters

We propose Algorithm 2, which only tries to match a best-response locally. We introduce a "current" hyperparameter $\hat{\lambda}$, which is updated each iteration. We define a conditional hyperparameter distribution, $p(\lambda|\hat{\lambda})$, which only puts mass close to $\hat{\lambda}$. Algorithm 2 combines the two phases of Algorithm 1 into one.

## 3 Related Work

**Model-free and Model-based approaches**   Simple model-free approaches applied to hyperparameter optimization include grid search and random search (Bergstra & Bengio, 2012). Hyperband (Li et al., 2016) combines bandit approaches with modeling the learning procedure.

Model-based approaches try to build a surrogate function, which can allow gradient-based optimization or active learning. An example is Freeze-thaw Bayesian optimization (Swersky et al., 2014) which can condition on partially-optimized model performance. Our work is complementary to the model-based SMASH algorithm of Brock et al. (2017), with section 2 discussing our differences.

**Optimization-based approaches**   Another line of related work attempts to directly approximate gradients of the validation loss with respect to hyperparameters. Maclaurin et al. (2015) differentiate through entire unrolled learning procedures. HOAG (Pedregosa, 2016) finds hyperparameter gradients with implicit differentiation by deriving an implicit equation for the gradient with optimality conditions. Franceschi et al. (2017) study forward and reverse-mode differentiation for constructing hyperparameter gradients.

## 4 Experiments

In all experiments, Algorithms 1 or 2 are used to optimize weights with a mean squared error on MNIST (LeCun et al., 1998) with $\mathcal{L}_{\text{Reg}}$ as an $L_2$ weight decay penalty weighted by $\exp(\lambda)$. The elementary model has $7,850$ weights. We used Adam for training the hypernet and hyperparameters.

### 4.1 Learning a global best-response

Our first experiment, shown in Figure 1, demonstrates learning a global approximation to a best-response function using Algorithm 1. We use 10 training data points to exacerbate overfitting. When training the hypernetwork, hyperparameters were sampled from a broad Gaussian distribution: $p(\lambda) = \mathcal{N}(0, 1.5)$. The minimum of the best-response in Figure 1 is close to the real minimum of the validation loss, which shows a hypernet can satisfactorily approximate a global best-response function in small problems.

### 4.2 Learning a local best-response

Figure 2 shows the same experiment, but using the Algorithm 2. The fused updates result in finding a best-response approximation whose minimum is the actual minimum faster than the prior experiment. The conditional hyperparameter distribution is given by $p(\lambda|\hat{\lambda}) = \mathcal{N}(\hat{\lambda}, 0.00001)$. This experiment shows that using only a locally-trained linear best-response function can give sufficient gradient information to optimize hyperparameters on a small problem. Algorithm 2 is also less computationally expensive than 1.

## 5 Conclusions and Future Work

In this paper, we addressed the question of tuning hyperparameters using gradient-based optimization, by replacing the training optimization loop by a differentiable hypernetwork. We gave a theoretical justification that sufficiently large networks will learn the best-response for all hyperparameters viewed in training. We also presented a simpler and more scalable method that jointly optimizes both hyperparameters and hypernet weights, allowing our method to work with manageably-sized hypernetworks.

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
