# OpenReview forum: "Stochastic Hyperparameter Optimization through Hypernetworks"
_ICLR.cc/2018/Workshop — Reject_

### Official Review · AnonReviewer1 · 2018-03-08
**Too packed, does not really explain the method**

**Rating:** 5
**Confidence:** 4

**Review:**

This paper is about a very interesting topic, but unfortunately, it does not work well in the workshop format -- the 3-pager simply appears to be too packed and doesn't leave me with an understanding of the proposed method.
The original conference submission was not invited to the workshop track (in which case there wouldn't even be a need for review, but the authors could just use the original submission).

Unfortunately, condensing the paper to 3 pages did not make it clearer. Section 2 claims to prove something about Algorithm 1, but the theorem statement only is about the existence of \phi^* and does not pertain to Algorithm 1. Furthermore, the theorem relies on assumptions that are not contained in the theorem statement, but only in its proof, e.g., that w_\phi is a universal function approximator (which, as stated in the reviews of the conference version, only holds in the limit). The proof is also not clear to me, e.g., the step "to satisfy Jensen's inequiality". Rather than including this incomplete proof (which necessarily remains unclear given the space constraints), in a workshop submission I believe it would have been far better to explain the intuition behind Algorithm 1 and Algorithm 2 and relate & compare them to known methods.

As a reviewer of the conference submission already pointed out, the authors are treating argmin as a single element, which is actually a problem here. This is dirty notation that is unfortunately far too wide-spread in the ML community, but it is even more unfortunate that the authors did not change notation in response to the reviewer's comment (that reviewer was not me); simply writing w* \in argmin_w [...] instead of w* = argmin_w [...] would really not be harder to understand and much cleaner. Writing \in would also make it clearer that right after Equation 2, "this function" is imprecise.

The abstract ends with "We compare this method to standard hyperparameter optimization strategies.", but there is no empirical comparison. There is a section called "related work", but that section just lists a couple of methods, without stating differences/similarities, except in the case of the very closely related SMASH approach (which is not a standard hyperparameter optimization strategies). I think the paper would either have to empirically compare or drop that claim from the abstract.

The paper does not appropriately reflect the limitations of the method, as e.g., stated in the reply to the review of the conference submission: "A good point is raised in that there are hyperparameters this algorithm can not optimize. We can not optimize hyperparameters about optimization, because there is no inner optimization loop."

When I read the second paragraph, I was confused that this paper claims to propose to learn this function, since I thought the SMASH paper proposed this; I think the late mention of SMASH on page 2 is a bit unfortunate.

Do I understand correctly that the experiments used MNIST with 10 training examples? Unfortunately, I cannot follow how the performed experiments demonstrate that the algorithm can be used for hyperparameter optimization. They don't even describe which hyperparameters are being optimized. The figures appear very useful, but without context I did not fully understand them.

Overall, I believe that there is something interesting here, but I cannot extract it from this too packed 3-pager. Nevertheless, I strongly encourage the authors to continue this line of work and the full version of the paper.

---

### Official Review · AnonReviewer2 · 2018-03-10
**no title**

**Rating:** 5
**Confidence:** 3

**Review:**

[ Paper Summary ]

The paper proposes a simultaneous joint optimization of a parameter and a hyper-parameter in a network, in which ``hypernet'' learns a functional relation from hyper-parameter to network parameters.

- novelty

The approach seems to be novel, though the topic is not my expertise.

- clarity

The basic idea is clear.

- significance

The problem setting would be significant. Technical significance of the proposed method might be slightly weak.

- quality

The presentation of experimental results would be able to be improved.

[ Comments ]

- pros

Automatic tuning of network would be useful.

The method is simple to implement.

- cons

Experiments are not convincing. The efficiency of the proposed method is not clear. Results (Figure 1 and 2) are not enough to see effectiveness.

Convergence of the algorithm 2 is not clear. In my understanding, Theorem 2.1 does not guarantee the convergence of the algorithm 2.

hyper-net would also have hyper-parameters.

Discussion about hyper-parameters other than lambda is missing.

---

### Decision · Program_Chairs · 2018-03-20
**ICLR 2018 Workshop Acceptance Decision**

**Decision:**

Reject

**Comment:**

Based on the reviews, this paper has not been accepted for presentation at the ICLR workshop. However, the conversation and updates can continue to appear here on OpenReview.